# A phylogenetic study of the members of the MAPK and MEK families across Viridiplantae

José Manuel González-Coronel[1], Gustavo Rodríguez-Alonso[1,2], Ángel Arturo Guevara-García[1]*

1 Departamento de Biología Molecular de Plantas, Instituto de Biotecnología, Universidad Nacional Autónoma de México, Cuernavaca, Morelos, México, 2 Centro de Investigación en Dinámica Celular, Instituto de Investigación en Ciencias Básicas y Aplicadas, Universidad Autónoma del Estado de Morelos, Cuernavaca, Morelos, México

* arturo.guevara@ibt.unam.mx

**Data Availability Statement:** All relevant data are within the manuscript and its Supporting Information files.

**Funding:** This work was supported by UNAM-DGAPA-PAPIIT (grant IN209420 to AAGG),

## Abstract

Protein phosphorylation is regulated by the activity of enzymes generically known as kinases. One of those kinases is Mitogen-Activated Protein Kinases (MAPK), which operate through a phosphorylation cascade conformed by members from three related protein kinase families namely MAPK kinase kinase (MEKK), MAPK kinase (MEK), and MAPK; these three acts hierarchically. Establishing the evolution of these proteins in the plant king-dom is an interesting but complicated task because the current MAPK, MAPKK, and MAPKKK subfamilies arose from duplications and subsequent sub-functionalization during the early stage of the emergence of Viridiplantae. Here, an *in silico* genomic analysis was performed on 18 different plant species, which resulted in the identification of 96 genes not previously annotated as components of the MAPK (70) and MEK (26) families. Interestingly, a deeper analysis of the sequences encoded by such genes revealed the existence of puta-tive domains not previously described as signatures of MAPK and MEK kinases. Addition-ally, our analysis also suggests the presence of conserved activation motifs besides the canonical TEY and TDY domains, which characterize the MAPK family.

## Introduction

Plants have evolved diverse and complex response mechanisms to contend against constantly changing environmental conditions. The perception of these conditions, together with the subsequent transduction and amplification of the generated signals, triggers cellular responses crucial to achieve optimal growth and development. Post-translational modifications (PTMs) constitute a major regulatory mechanism of protein activity [1, 2]. PTMs regulate the activa-tion or inhibition of protein activity, change the subcellular localization, alter protein stability, and promote or prevent trans-interactions [1–3]. PTMs are catalyzed by a wide variety of enzymes and are usually reversible [2, 4, 5]. Particularly, protein kinases regulate processes at the transcriptional, translational and post-translational level by catalyzing the addition of a phosphoryl group ($PO_3^{2-}$) from ATP to a substrate protein in specific amino acid residues usu-ally serine (S), threonine (T) or tyrosine Y 6–8. Phosphorylation reactions depend on the

PRODEP (postdoctoral fellowship granted to GRA), and CONACYT-México (fellowship 481085 to JMGC). The funders had no role in study design, data collection and analysis, decision to publish, or preparation of the manuscript.

**Competing interests:** All the authors have declared that no competing interests exist.

catalytic activity of the kinase and the affinity, or protein-protein interaction capability, for their target proteins [5, 6].

Versus other eukaryotic genomes, plants contain a larger number of protein kinases; for example, the *Arabidopsis thaliana* L. genome encodes approximately 1,200 kinases while the rice (*Oryza sativa* L.) genome contains ~1,400 kinase genes [7, 8]. On the other hand, approximately 500 and 120 protein kinases are encoded in the *Homo sapiens* L. and *Saccharomyces cerevisiae* Meyen ex E.C. Hansen. genomes, respectively [9].

The MAPKs belong to the group of serine/threonine protein kinases that are responsible for transforming extracellular *stimuli* into a wide range of cellular responses [5, 10]. MAPK signaling cascades are highly conserved in all eukaryotic organisms and are made up of three different gene families MAPK or MPK, MAPK kinase (MKK or MEK), and MAPKK kinases (MPKKK or MEKK) [5, 7, 11]. These components are sequentially activated by phosphorylation in the activation domain of their substrates [12, 13]. The first component of the signaling cascade is the MEKK; it phosphorylates a pair of serine (S) and threonine (T) residues in the S/T-X$_{3-5}$-S/T domain of MEK proteins. Subsequently, MEKs activate MAPKs by phosphorylation of threonine (T) and tyrosine (Y) residues in the activation domain (T-X-Y) [2, 14, 15].

The modular and hierarchical arrangement of the MAPK signaling pathways is suitable for amplification and integration of signals at the cellular level [16, 17]. Signaling cascades mediated by MAPKs coexist in many cells and are connected and regulated by feedback to at least some degree [2, 10, 11, 16, 18, 19]. Due to the central role played by the MAP kinases in signal transduction and transmission, it is of great interest to classify the members of these families based on their phylogenetic relationships and their functional characteristics [16, 20]. Despite the growing availability of plant genomes, the identification of MPKs has been limited to a small number of species, mainly from monocots [21–23], and dicots [12, 14, 24–30].

The *Arabidopsis thaliana* L. genome contains 20 genes encoding MAPKs, 10 for MEK, and 80 for MEKK. The number of MEK genes is usually much smaller than that of the MAPK and MEKK families [14, 19, 31, 32]. MEKK proteins represent the largest family of signaling cascade components mediated by MAP kinases with 80 members classified into three groups: MAPKKK, ZIK, and RAF consisting of 21, 11, and 48 genes respectively [19]. Members of the MAP kinase family have 11 canonical domains with an activation domain between domains VII and VIII. The activation domain of MAP kinases contains a pair of threonine and tyrosine residues, which are phosphorylated by a MEK protein [33]. Angiosperm MAPKs are classified into four groups (A-D) on the basis of their activation domain [14].

The molecular activity of MAP kinases has been seen in *Arabidopsis*. For example, the MEKK1-MKK4/5-MPK3/6 module participates in flagellin-triggered immune response [34]; and the MEKK1-MEK1/2-MPK4/6 module is activated in response to different types of stress [35].The module MEKK4-MEK4/5-MPK3/6 was initially identified as a regulator of stomata development [36], and later suggested to participate in the regulation of root and embryo development [37]. In this work, MAP kinases from seven species not previously analyzed were identified: *Amaranthus hypochondriacus* L., *Azolla filiculoides* [Lam.], *Isoetes echinospora* Durieu, *Marchantia polymorpha* L., *Pinus taeda* L., and *Ostreococcus tauri* C. Courties & M.-J. Chrétiennot-Dinet, *Salvinia cucullata* [Bory.]; to provide a list of members of the MEK and MAPK families of each of the 18 analyzed species. A total of 70 and 26 novel sequences of MAPK and MEK proteins were obtained, respectively. In addition to a detailed study of the members of the MAP kinase family of the aforementioned species, a comparative analysis was also perfomed by adding the genes that code for the members of such families in representative species of the different plant lineage, including a chlorophyta (*Chlamydomonas reinhardtii* PA Dang.), a lycophyte (*Selaginella moellendorffii* P. Beauv.), two bryophytes (*Physcomitrella patens* [Hedw.] Bruch & Schimp., and *Sphagnum fallax* H. Klinggr.), a gymnosperm (*Picea abies*

[L..] H. Karst..), and four angiosperm species: *Vitis vinifera* L., *Beta vulgaris* L., *Brachypodium distachyon* [L.] P. Beauv. and *Amborella trichopoda* Baill.

## Materials and methods

### Identification of MAPK and MEK genes in Viridiplantae

A BLASTP was carried out using the sequences of the MAPK and MEK proteins of *Arabidopsis thaliana* L. as a query to find putative orthologs in *Amaranthus hypochondriacus* L., *Amborella trichopoda* Baill., *Chlamydomonas reindhardtii* P.A. Daung., *Physcomitrella patens* [Hedw.] Bruch & Schimp., *Selaginella moellendorffii* P. Beauv., *Sphagnum fallax* H. Klinggr. (https://phytozome.jgi.doe.gov/pz/portal.html), *Azolla filiculoides* [Lam.], and *Salvinia cucullata* [Bory.] (https://www.fernbase.org/), *Marchantia polymorpha* L. (https://marchantia.info), *Oryza sativa* L. (http://www.plantgdb.org/OsGDB/), *Ostreococcus tauri* C. Courties & M.-J. Chrétiennot-Dinet. (https://genome.jgi.doe.gov/Ostta4/Ostta4.home.html), *Pinus taeda* L., *Picea abies* [L.] H. Karst. (http://congenie.org/), and *Vitis vinífera* L. (http://www.genoscope.cns.fr/externe/GenomeBrowser/Vitis/) databases; considering an expected value (e-value) of $1x10^{-35}$ as a threshold. An additional search was carried out using hidden Markov models using HMMER 3.0 software [38]. Subsequently, the presence of the serine/threonine protein kinase domain (PF00069) was corroborated in the retrieved sequences.

Those proteins containing the D-[L/I/V]-K and S/T-$X_{3-5}$-S/T sequences were considered putative MEK proteins [14, 21]. The putative members of the MAPK family must contain the characteristic sequence [L/I/V/M]-[TS]-X-X-[L/I/V/M]-X-T-[K/R]-[W/Y]-Y-R-X-P-X-[L/I/V/M]-[L/I/V/M] including the T-X-Y activation domain [14, 28]. The NCBI Conserved Domain Database (http://blast.ncbi.nlm.nih.go), ProtParam online software from the ExPASy suite (https://web.expasy.org/protparam/), and the InterProScan database (https://www.ebi.ac.uk/interpro/search/sequence-search) were used to validate the presence of domains in each of the sequences. Finally, the molecular weight (MW) and the isoelectric point (pI) of each protein was predicted with the ExPASy server (http://web.expasy.org/compute_pi/). Boxplots for molecular weight and isoelectric point were visualized using PlotsOfData software (https://huygens.science.uva.nl/PlotsOfData/).

### Multiple sequence alignment, phylogenetic, and gene structure analysis

Multiple sequence alignment was generated using the iterative refined method E-INS-i from MAFFT online software (Multiple Alignment using Fast Fourier Transform; https://mafft.cbrc.jp/alignment/server/) [39], (S1 and S2 Files) and visualized with Jalview software v2.11.1.3 [40]. The IQtree 1.6.6 (http://www.iqtree.org/) and ProtTest (http://darwin.uvigo.es) software were used to determine the evolution substitution model of proteins to be used for the construction of the phylogenetic tree [41, 42]. The phylogenetic analysis was performed using the complete sequence of each protein with the IQtree v1.6.6 software. The maximum-likelihood algorithm was used. The tree topology was statistically tested with the bootstrap method with 1,000 iterations (S3 and S4 Files). The tree topology was visualized in the iTOL software v5.7 [43].

### Identification of novel domains and protein structure prediction

MEME-suite software v5.2.0 [44] was used to identify the canonical and novel MAPK and MEK domain. MEME was run with default parameters and a predefined motif length of eight to 12–15 residues. PhosphoSVM software [45] was used to predict phosphorylation of the novel domains. The protein logos were performed with WebLogo software v2.8.2 [46].

The protein structure was predicted with the I-Tasser software v5.1 [47] and visualized in the PyMol software v2.4.1 (The PyMOL Molecular Graphics System, Version 2.0 Schrödinger, LLC.). The MPK6 crystal structure (PDB id: 5ci6) [48] was used as a template for structure modelling. Figure composition was performed using the InkScape software v1.0 (https://inkscape.org/es/).

## Results and discussion

### Identification and nomenclature

MAP kinases are of paramount importance for plant development and stress response [2]. The partial redundancy of this protein superfamily, the high sequence similarity among homologs, and the expansion of the gene family in diverse plant species, have hindered the evolutionary studies of MAPKs. To provide a comprehensive phylogenetic analysis of MAPK proteins, a genome-wide identification of MAPK genes was performed for 18 selected species, spanning the major clades of Viridiplantae. Our analysis includes chlorophyta (*C. reindhardtii*, and *O. tauri*), bryophytes (*M. polymoprpha*, *P. patens*, and *S. fallax*), lycophyte (*S. moellendorffii*), seedless vascular plants (*A. filiculoides*, *I. echinospora*, and *S. cucullata*), gymnosperms (*P. taeda* and *P. abies*) and angiosperms. For the last clade we include the basal angiosperm *Amborella trichopoda*, as well as monocotyledonous (*B. dystachion*, and *O. sativa*) and dicoty-ledoneous species (*A. hypochondriacus*, *A. thaliana*, *B. vulgaris*, and *V. vinifera*). MAPK genes were identified through a combined approach of homology identification and hidden Markov models (Table 1).

A total of 249 genes were identified: 153 previously described and 96 not previously anno-tated as members of any of the three different modules that constitute a canonical MAP kinase cascade (S1 Table); 70 of the novel MAPK genes belong to MAPK and 26 to the MEK subfam-ily, respectively. Although previous efforts to reconstruct the evolutionary history of MAPK genes have been made, ortholog identification on early divergent plant clades has remained elusive [12]. Moreover, independent studies in species such as *A. thaliana*, *O. sativa*, and *P. tri-chocarpa*, has led to different nomenclature and classification systems for MAPKs [14].

Here, using MAPKs as an example, a nomenclature system was stablished in which each of the genes were named using a two-letter code corresponding to the first letter from the genus and species. This two-letter code remained in: Af (*Azolla filiculoides*), Ah (*Amaranthus hypo-chondriacus*), At (*Arabidopsis thaliana*), Bv (*Beta vulgaris*), Bd (*Brachypodium distachyon*), Cr (*Chlamydomonas reindhardtii*), Ie (*Isoetes echinospora*), Mp (*Marchantia polymorpha*), Ot (*Ostreoccocus tauri*), Os (*Oryza sativa*), Pp (*Physcomitrella patens*), Pa (*Picea abies*), Pt (*Pinus taeda*), Sc (*Salvinia cucullata*), Sf (*Sphagnum fallax*), Sm (*Selaginella moellendorffii*), and Vv (*Vitis vinífera*); to distinguish *Amborella trichopoda* from *A. thaliana*, the second letter of both genus and species was added to the first one (i.e., Amtr). Next to this letter code the acronym MPK (from **M**itogen-activated **P**rotein **K**inase) or MEK (MAPK/ERK kinase) was included along with a and a number referring to its most likely ortholog in *Arabidopsis thaliana* L [49]. Likewise, when two or more of the identified sequences have the same putative ortholog in *Arabidopsis*, they were distinguished by adding a letter in alphabetical order. This is the case of the *P. taeda* sequences PITA_000030510, PITA_000007088 and PITA_000001460, which, were putative orthologs of AtMPK5 and were renamed as PtMPK5a, PtMPK5b and PtMPK5c, respectively.

### Phylogenetic analysis of MAP kinases

A multiple sequence alignment of the retrieved MAP kinases sequences was built and used to reconstruct the molecular phylogeny of MAP kinases. The alignment shows a high degree of

**Table 1. Number of MAPK genes present per genome (specie).**

| | Specie | Lineage | MAPKs gene number |
|---|---|---|---|
| 1 | *Arabidopsis thaliana* L. | Angiosperm (Eudicot) | 20 |
| 2 | *Amaranthus hypochondriacus* L. | Angiosperm (Eudicot) | 12 |
| 3 | *Beta vulgaris* L. | Angiosperm (Eudicot) | 7 |
| 4 | *Vitis vinifera* L. | Angiosperm (Eudicot) | 14 |
| 5 | *Brachypodium distachyon* [L.] P. Beauv. | Angiosperm (Monocot) | 14 |
| 6 | *Oryza sativa* L. | Angiosperm (Monocot) | 15 |
| 7 | *Amborella trichopoda* Baill. | Angiosperm (Basal) | 8 |
| 8 | *Picea abies* [L.] H. Karst.. | Gymnosperm | 11 |
| 9 | *Pinus taeda* L. | Gymnosperm | 12 |
| 10 | *Azolla filiculoides* [Lam.] | Pteridophyta | 15 |
| 11 | *Salvinia cucullata* [Bory.] | Pteridophyta | 14 |
| 12 | *Isoetes echinospora* Durieu. | Lycophyta | 8 |
| 13 | *Selaginella moellendorffii* | Lycophyta | 5 |
| 14 | *Sphagnum fallax* H. Klinggr. | Bryophyta | 8 |
| 15 | *Marchantia polymorpha* L. | Bryophyta | 6 |
| 16 | *Physcomitrella patens* [Hedw.] Bruch & Schimp. | Bryophyta | 11 |
| 17 | *Chlamydomonas reindhardtii* P. A. Dang. | Algae | 6 |
| 18 | *Ostreococcus tauri* C. Courties & M.-J. Chrétiennot-Dinet | Algae | 3 |

conservation in the sequences that correspond to the 11 characteristic MAPK domains; moreover, all analyzed sequences contain the TXY-activation domain (S1 Fig). A maximum likelihood tree was constructed for the MAP kinase-retrieved sequences; the tree was rooted with the *Saccharomyces cerevisiae* Fus3 protein that was selected as an outgroup given its similarity to Viridiplantae MAPKs. Previous studies classified the *Arabidopsis* MAPK genes into four groups (A, B, C and D) according to their sequence similarity and the presence of the TDY or TEY phosphorylation motifs [14]. The resulting ML tree topology displays five well-supported clades with bootstrapping values >90.0% out of 1000 replicates; thus, we support the previous suggestion to consider another group to classify MAP kinases (A-E); the existence of an additional group F has been refuted [50] (Fig 1).

The MAPK group A contains sequences retrieved from pteridophytes, gymnosperms and angiosperms. Surprisingly, no group A MAPKs were identified in the lycophytes, bryophytes or algae species analyzed. This observation suggests that group A MAPKs arose in the Euphyllophyte clade after the separation from the lycophytes, which occurred ca. 420 million years ago (MYA) [51, 52] (Figs 1 and 2; S1 Table). AtMPK3 and AtMPK6 both belong to group A MAPKs and are the two most widely studied MAPK proteins. Together with their putative orthologs in other plant species, these species have been involved in responses to biotic and abiotic stresses [19, 53, 54]. Members of group B have been involved in cell division and in the responses to biotic and abiotic stress; in particular, the loss of function mutant *mpk4* of *A. thaliana* leads to a constitutive phenotype of systemic acquired resistance [55, 56]. Some members such as AtMPK13, another group B MAPK, are activated through the cell cycle, and they are located specifically in the phragmoplast during telophase [57].

Groups A and B are sister clades according to the tree topology; this observation together with its absence in algae species suggests that these MAPK groups were the last to be acquired (Figs 1 and 2). Group C MAPKs are less characterized than MAPKs from groups A and B, although the expression and activity of AtMPK7 (group C) is regulated by the circadian cycle [58]. Group D makes up the largest group of MAPK proteins including 61 sequences. This is

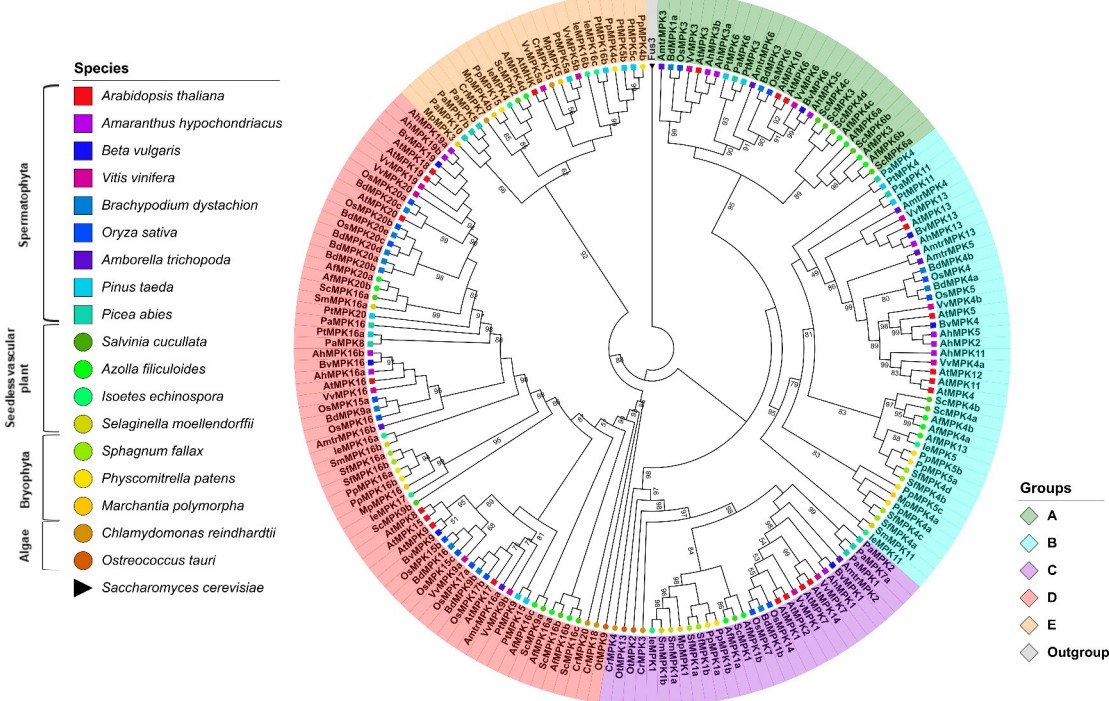

**Fig 1. Phylogenetic tree of MAPKs proteins in Viridiplantae.** Maximum-likelihood tree reconstructed from amino acid sequences. The tree was rooted with the Fus3 protein from *Saccharomyces cerevisiae*. Bootstrap values from 1,000 replicates are indicated at nodes with support values >49%. Each MAPK group is indicated with a distinctive color. Different species are indicated with shapes and colors.

characterized by the sequence TDY in its activation domain (S2 Fig); in addition, group D MAPKs have an extended C-end compared to proteins from the groups A-C. In *Arabidopsis*, the AtMPK8 protein from group D is involved in seed germination and dormancy [59].

Finally, members of group E are known as **M**apk-**h**omologous **k**inases (MHKs) and may contain the TEY or TDY sequence and even some non-canonical motif in the activation domain (S2 Fig) [14]. It remains a matter of debate whether group E MAPKs constitute functional MAPKs given their similarity to cyclin-dependent kinases (CDKs), even though their biochemical properties are largely unknown.

**MAPK analysis.** Of the genomes under study, the angiosperm clade has the highest number of MAPK members per family, with *A. thaliana* having the highest number of genes encoding MAPKs (20) followed by rice (*O. sativa*) and grape (*Vitis vinifera*) with 15 and 14 genes, respectively. In addition, species *A. filiculoides* and *S. cucullata* in the pteridophytes clade have 15 and 14 genes, respectively. The large number of MAPK genes in Azolla and Salvinia might be a result of whole genome duplication events in the Salviniales [60]. On the other hand, *O. tauri* in the algae group has the lowest number of genes of the analyzed species (3), while *C. reindhardtti* has 6. Bryophytes have an increase in the number of MAPK genes with respect to that of algae. The members of this clade, *P. patens*, *S. fallax*, and *M. polymorpha* contain 11, 8, and 6 genes, respectively. The MAPK gene family expansion in bryophytes is congruent with the expansion of gene families in land plants upon terrestrialization and might be associates with plant facing new types of stress, such as dehydration, gravity, exposure to ultraviolet light, etc. after land colonization [61, 62] (Fig 1; Table 1).

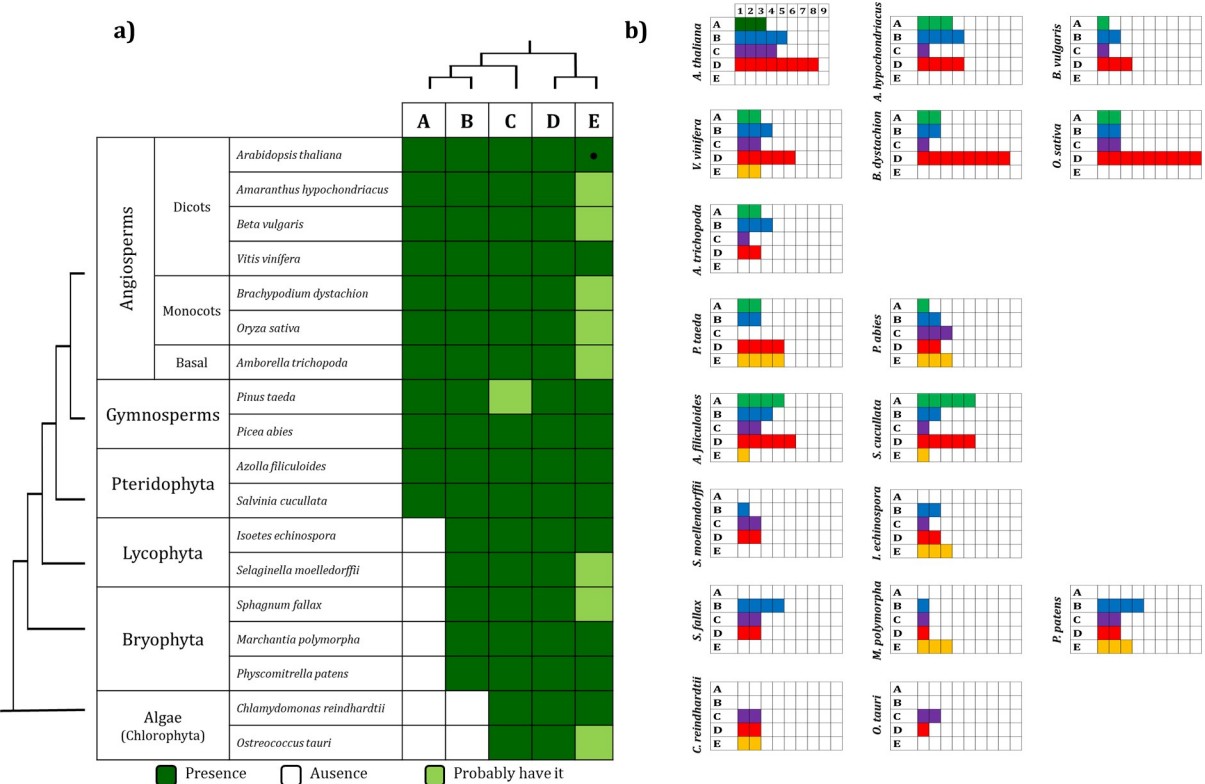

**Fig 2. Scheme representing the number of MAPKs present in each group.** a) Presence of the MAPK groups across Viridiplantae species. b) Number of MAPKs in each group in each species analyzed. Group E MAPKs, including AtMHK (•) were considered in the analysis although its recognition as a MAPK subfamily is still debated (see text). Colors in b) refer to the color used in Fig 1.

Among the 162 MAPK proteins identified, those belonging to group D are the longest. VvMPK7 (GSVIVT01018883001) of *V. vinifera* is the longest protein with 770 amino acids. In general, MAPKs proteins that belong to organisms from higher taxa such as angiosperms and gymnosperms have longer sequences than organisms of lower taxa except CrMPK18 (Cre17. g745447) of the alga *C. reinhardtii* with a length of 765 amino acids. On the other hand, the PaMPK2 protein from *Picea abies* has the shortest length (172 amino acids). Regarding the structure of the MAPK gene, there seems to be no specific pattern of intron organization in terms of the taxonomic level to which each species belongs including genes with zero (6 sequences), one (18), two (8), three (5), four (5), five (46), six (8), seven (10), eight (8), nine (33), ten (20), eleven (2), twelve (2), fourteen (2), fifteen (7) and sixteen (1) introns. The MpMPK3 gene of *M. polymorpha* contains the maximum number of introns (16), and none of the three genes identified in the unicellular algae *O. tauri* have introns. Furthermore, 18 of the sequences listed above contain at least one intron in the 3' or 5' UTR (**Un**T**ranslated **R**egion) according to their annotated gene models (S1 Table). Regarding the organization of the groups, however, those proteins that belong to groups D and E have the highest number of introns while the members of group C have the least amount. Notably, gymnosperm and angiosperms organisms analyzed contain at least one intron. The predicted molecular weights of each of the MAP kinases reported here range from 20,559 Da (PaMPK2) to 87,480 Da (VvMPK7); while their isoelectric points (Ip) range from 4.66 (PaMPK2) to 9.72 (AfMPK16a) (S1 Table). Interestingly, group D and E comprise the largest proteins, with an average peptide

length of 558 and 461, respectively (63,358 and 52,287 Da); these groups contain the most basic theorical isoelectric points averaging 8.45 and 8.10, respectively (S3 Fig).

**Analysis of motifs and domains conserved in the MAPK family.** MAPKs are characterized by the presence of multiple domains. Eight domains, named I-V, VIa, VIb, and VII are distributed towards the N-terminal preceding the activation domain. Towards the C-terminal end of the activation domain, there are five other domains previously described as typical of MAPKs. Domain VIII, IX, and X precede the CD domain; finally, the XI domain is localized around 30 amino acids after the CD domain (Table 2) [33, 35]. We used the MEME's bioinformatics software to address domain conservation among the identified MAPK sequences [44].

Of the 90 sequences analyzed that belong to angiosperms, 83 present all the canonical domains. The remaining seven (2 sequences from *A. trichopoda* and 5 from *V. vinifera*) lack at least one domain: VvMPK1 and VvMPK7 (domains I-IV), VvMK5a and VvMPK5b (III-V and VIa), VvMPK20 (I -III), AmtrMPK5 (IV), and AmtrMPK16a (XI). The fact that the vast majority of sequences belonging to angiosperms contain all the canonical domains could reflect a bias because this clade is the most studied (S4 Fig; S2 Table). Most of the sequences of the analyzed angiosperm species lack multiple domains. In *P. taeda*: PtMPK5a (III-V, and VIa), PtMPK5b (III-V, VIa, and IX), PtMPK5c (III-V, VIa, and IX-X), PtMPK11 (I -III), PtMPK16 (II-V, and VIa), and PtMPK20 (IX-X). In *P. abies*: we note PaMPK2 (I-VII), PaMPK4a (V), PaMPK7a (VIb and VII-XI), PaMPK7b and PaMPK10 (III, V, and VIa), PaMPK11 (I-III, and V), and PaMPK16 (IX and XI). Eight of these sequences belong to group E MAPKs, which include MHKs and whose inclusion as MAPKs, as already mentioned, has been debated due to their sequence similarity to CDKs [11]. Whether group E proteins function as MAPKs has been poorly studied, although an ortholog of these proteins in the fungus *Ustilago maydis* is activated by phosphorylation by a MAPKK and contains the activation motif TXY [63]. This observation supports the hypothesis that group E MAPKs should be regarded as a subfamily of MAPKs. In fern species, the AfMPK6b and AfMPK16b sequences of *A. filiculoides* lack domain I and domain IX, respectively. The AfMPK4d sequence lacks domains IV and IX. On the other hand, in *S. cucullata*, the ScMPK6b sequence lacks the V, VIa and VIb domains; the ScMPK16c sequence lacks domain XI. Domains III, IV and VII are absent in the ScMPK2 sequence. Finally, domain I is absent in the ScMPK3, ScMPK4a, and ScMPK4b sequences. In the case of *I. echinospora*, the sequence IeMPK4 lacks I-V domains while IeMPK7 and IeMPK11 lacks domain XI (S4 Fig; S2 Table).

**Table 2. Consensus sequences of the conserved domains in MAPKs.**

| Domain | Consensus sequence |
|---|---|
| I | [V/P]-[I/V]-G-[K/R]-G-[S/A]-Y-G-[V/I]-V-C-S-A |
| II | E-X-V-A-I-K-K-I-X-[N/D]-[A/V/I]-F-[E/D]-[N/H]-$X_2$-D-A |
| III | R-[T/I]-L-R-E-[I/L]-K-L-L-R-[H/L]-[L/M]-[R/D] |
| IV | P-X-[R/K]-$X_2$-F-X-D-[V/I]-Y |
| V | V-[F/Y]-E-L-M-[E/D]-[T/S]-D-L-H-Q-[V/I]-I-[K/R] |
| VIa | [F/Y]-F-L-Y-Q-[L/I/M]-L-R-[G/A]-L-K-Y |
| VIb | H-[S/T]-A-N-[V/I]-[L/F/Y]-H-R-D-K-L-P-[K/S]-N-[L/I]-L-[A/L]-N |
| VII | C-[D/K]-L-K-I-[C/A]-D-F-G-L-A-R-[V/T] |
| VIII | [V/A]-T-R-W-Y-R-A-P-E-L-[L/C]-[L/G]-[S/N] |
| IX | A-I-D-[I/V/M]-W-S-[V/I]-G-C-I-F-[A/M]-E-[L/I/M]-[L/M] |
| X | P-[L/I]-F-P-G-$X_3$-[V/L]-X-Q-L-X-L-[I/M]-T-[D/E] |
| XI | F-D-P-$X_2$-R-[I/P]-[T/S]-[A/V]-X-[E/D]-A-L-X-[H/D]-P-Y-[F/L] |
| CD | L-H-D-$X_2$-D-E-P |

The lack of several domains in the angiosperm and fern sequences is due to the fact that the genomes of these species are still fragmented, in the databases. Some of the ORFs that are found as hits might come from fragmented sequences and therefore the sequences corresponding to certain domains are missing. In the bryophytes *M. polymorpha*, *S. fallax*, and *P. patens*, we see MpMPK3 and MpMPK5 (III-V, and VIa), MpMPK4b (III and IV), PpMPK4b (II-IV, VIa, and IX), PpMPK4c (III-V, VIa, and IX), PpMPK15 (III, IV, VIa, and IX), PpMPK16a, and PpMPK16b (IX). For their part, the three sequences identified here from the alga *O. tauri* lack domain IV. Thus, the results of this analysis allow us to suggest that the presence of the activation motif (T-X-Y) together with five domains (II, III, V, and VII) located towards the N-terminal end of the activation domain and present in all the analyzed sequences are sufficient to identify MAP kinases from any plant species. In addition to this, the presence of domains VIII-XI in most Viridiplantae MAPK sequences suggests that these domains are of critical importance for maintaining either the function or structure of these proteins (S4 Fig; S2 Table).

MAPK proteins contain a T-X-Y motif whose phosphorylation leads to MAPK activation [14]. The Viridiplantae multiple sequence alignment shows that all analyzed species contain the canonical activation domain in the form of T-E-Y (98 sequences), T-D-Y (77 sequences), and T-S-Y (1 sequence). The alignment also shows the presence of non-canonical activation domains including M-E-Y (2 sequences), T-E-M (2 sequences), T-H-E (4 sequences), T-H-L (1 sequence), T-H-Q (1 sequence), T-K-T (2 sequences), T-Q-M (1 sequence), and T-S-Y (1 sequence). The distribution of these activation domains on the MAPK phylogeny shows that the T-E-Y domain is present in MAPKs from groups A, B, C, and E; while the T-D-Y is present on the D and E groups (S2 Fig; S2 Table). The activation domain contained in group E sequences exhibits the highest variability, which is especially evident on Bryophyta including T-D-Y, T-H-E, T-K-T, and T-Q-M variants (S2 Fig).

In addition to the 11 characteristic domains, MAP kinases also contain the sequence (L/H)-D-$X_2$-D-E-P known as the common docking domain (CD domain) [19]. Our results show that the CD domain is present in most proteins from the A, B, and C MAPK group; although it is also present in a few sequences from the groups D and E, it is generally absent in proteins from these clades (S4 Fig; S2 Table). The CD includes a motif involved in recognition and binding to MAPK substrates, as well as in protein-protein interaction with MEKs. The adjacent aspartate (D) and glutamate (E) residues in the CD, are essential for interaction with the basic residues lysine (K) and arginine (R), located in the binding site of MEK proteins [64]. It remains to be experimentally demonstrated whether the absence of the CD domain has any impact on the function of these group D and E MAPKs either for interaction and phosphorylation by MEKs or in MAPKs substrate specificity.

**Identification of novel and distinctive domains for each group of MAPKs.** The domain analysis of the retrieved sequences suggests the existence of six novel domains, named 12–17 (Fig 3; S3 Table). The presence of these domains might predict the group to which a MAPK protein belongs facilitating MAPK identification during genome or transcriptome annotation. The consensus sequences of such domains consist of 12–17 amino acid residues (Table 3).

Domain 12 is located approximately 15 residues before domain I from group A and B proteins. It is also present in OtMPK13 from *O. tauri*. Domain 13 is also located towards the N-terminus end of domain I but is only present in the proteins from group C therefore providing a signature for group C MAPKs. Domain 14 is located between domains II and III and typifies proteins from group D. Domains 15–17 are distributed towards the C-terminal end and, if present, they are always located before the CD domain between the canonical domains X and XI. Domain 15 is present in proteins from the B and E group with the sole exception of PtMPK16b from group E. Except VvMPK3 and VvMPK6 that belong to group A, domain 16

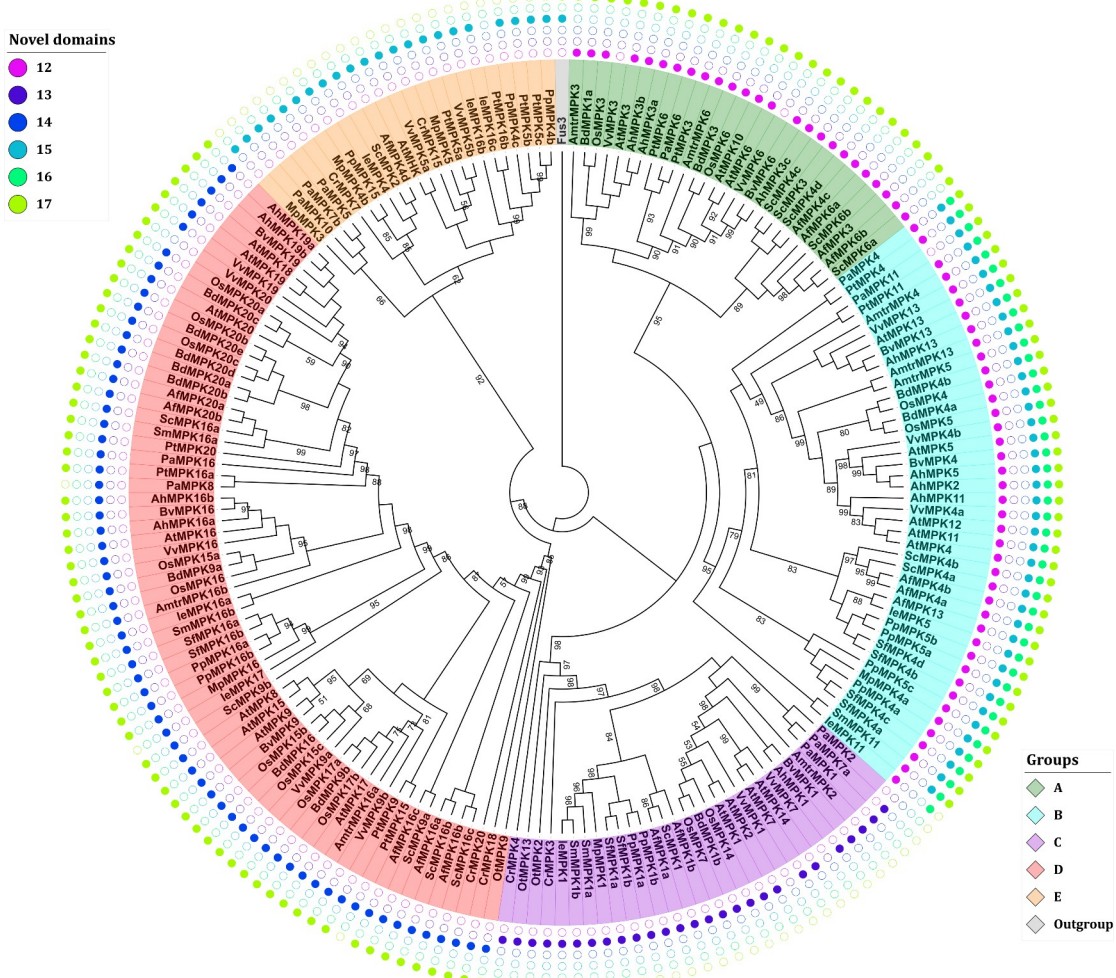

**Fig 3. Phylogenetic analysis and identification of novel domains in MAPKs.** The analysis of novel domains was performed with the MEME suite [44]. The distribution of the novel domains is group specific and could, therefore, facilitate MAPK identification and annotation. Sequence nomenclature follows a two-letter code to indicate genus and species (see text).

is only located in group B MAPKs. Domain 17 is present in most proteins except those in groups C and E. Interestingly, domain 17 is also present in the Fus3 kinase of *S. cerevisiae* this prompts the hypothesis that this domain was present in the ancestral MAPK sequences and was lost in groups C and E (Fig 3; S3 Table).

The conservation and distribution of these novel MAPK domains in a group-specific manner could suggest that they are involved in either maintaining the structure of MAPKs or in

**Table 3. Consensus sequences of the novel domains discovered in MAPKs.**

| Domain | Consensus sequence |
|---|---|
| 12 | G-N-X-F-E-V-[T/S]-X-K-Y |
| 13 | Y-X-[M/L]-W-[Q/R]-[T/S]-X-F-E-I-D-T-K-Y |
| 14 | H-[P/K]-D-I-V-E-[I/V/K]-[K/L]-[I/H/N]-[I/K]-[M/L]-L-P |
| 15 | M-L-X-F-[D/N]-P-$X_2$-R-I-[T/S] |
| 16 | E-[L/V]-[I/L]-G-[T/S]-P-X-[E/D]-X-D-L-X-F-[L/I/V] |
| 17 | A-[R/K]-[R/K]-Y-[L/I/V]-$X_2$-[L/M]-[R/P]-$X_3$-[P/R/K]-X-[P/S] |

fine-tuning the selectivity towards substrates or protein interactors. This hypothesis is supported by homology structure prediction of selected MAPKs, which shows that domains 12, 13, 15, 15, and 17 are exposed to the solvent and could therefore provide an interaction surface. Meanwhile domain 14 is in a pocket (S5 Fig). Moreover, a computational prediction of phosphorylation sites suggests that domains 15 and 16 contain amino acid residues susceptible to phosphorylation (S4 Table). Whether these residues are indeed phosphorylated or if the identified novel domains are of relevance for the MAPKs structure and function remains to be experimentally demonstrated and extends beyond the scope of this study.

## Identification of MEK proteins across Viridiplantae

Here, 98 MEK genes were identified in the analyzed species. *B. distachyon* was the species with the highest number of MEK genes (12) followed by *A. thaliana* and *O. sativa* with 10 and 9 genes, respectively. The algae *O. tauri* and *C. reindhardtii* are the species with the lowest number of MEK genes each having only one. The number of MEK genes encoded in the genome of the analyzed species suggests a gradual but constant gene MEK expansion in Viridiplantae (Table 4).

In general, the proteins that make up the MEK family have been characterized by a smaller size compared to those belonging to the MAPK family, having a length of ~350 amino acid residues. However, the longest protein of this group, PtMEK4 has 1333 amino acids. On the other hand, the smallest protein is PaMEK3 from *P. abies* with a length of 133 amino acids. The arrangement of introns presents in the MEK sequences range from 1 to 10. Unlike MAPK genes that usually contain one or more introns, 27 of the identified MEK genes (> 30%) lack introns. The MEK sequences do not show an intron arrangement that characterizes the gene family including genes with one (6 sequences); two (3); three (1); four (3); three (1); four (3); five (2); six (5); seven (17); eight (11); nine (9); and ten introns (1) (S1 Table). In addition, transcriptome analysis of *I. echinospora* show that four MEK sequences have been identified, but no information about the number of introns can be gathered from this data [49]. The

**Table 4. Number of MEK genes present per genome (species).**

| | Specie | Lineage | MEKs gene number |
|---|---|---|---|
| 1 | *Arabidopsis thaliana* L. | Angiosperm (Eudicot) | 10 |
| 2 | *Amaranthus hypochondriacus* L. | Angiosperm (Eudicot) | 5 |
| 3 | *Beta vulgaris* L. | Angiosperm (Eudicot) | 5 |
| 4 | *Vitis vinifera* L. | Angiosperm (Eudicot) | 5 |
| 5 | *Brachypodium distachyon* [L.] P. Beauv. | Angiosperm (Monocot) | 12 |
| 6 | *Oryza sativa* L. | Angiosperm (Monocot) | 9 |
| 7 | *Amborella trichopoda* Baill. | Angiosperm (Basal) | 7 |
| 8 | *Picea abies* [L.] H. Karst.. | Gymnosperm | 7 |
| 9 | *Pinus taeda* L. | Gymnosperm | 3 |
| 10 | *Azolla filiculoides* [Lam.] | Pteridophyta | 6 |
| 11 | *Salvinia cucullata* [Bory.] | Pteridophyta | 4 |
| 12 | *Isoetes echinospora* Durieu. | Lycophyta | 4 |
| 13 | *Selaginella moellendorffii* | Lycophyta | 3 |
| 14 | *Sphagnum fallax* H. Klinggr. | Bryophyta | 6 |
| 15 | *Marchantia polymorpha* L. | Bryophyta | 3 |
| 16 | *Physcomitrella patens* [Hedw.] Bruch & Schimp. | Bryophyta | 7 |
| 17 | *Chlamydomonas reindhardtii* P. A. Dang. | Algae | 1 |
| 18 | *Ostreococcus tauri* C. Courties & M.-J. Chrétiennot-Dinet | Algae | 1 |

molecular weight of the identified MEK proteins range from 5,877 Da (AhMEK3) to 74,002 Da (PtMEK4), while the isoelectric points range from 5.24 and 10.15 (S1 Table; S6 Fig).

**Phylogenetic analysis.** A multiple alignment of the retrieved MEK sequences was generated this showed that MEK proteins are highly similar and that the 8 characteristic domains of MEK proteins are highly conserved as well as the ATP-binding and MAPK-binding domains (S8 Fig). Ichimura *et al.* classified MEK proteins into four groups, but this classification system was generated from angiosperm sequences only [14]. To avoid bias in the phylogeny reconstruction sequences retrieved from gymnosperms, seedless vascular plants, and non-vascular plants were included as well as the MEK protein Ste7 from *S. cerevisiae* as an outgroup to root the tree. The obtained ML topology shows that MEK proteins partition into five groups named A-E (Fig 4).

MEK proteins are involved in a plethora of biological processes; for example, some members of group A, such as AtMEK1 and AtMEK2 are involved in responses to environmental stimuli [35, 65] while AtMEK6 participates in cell division [66]. The MEK proteins included in group B have an extended C-terminal with the conserved sequence E-[K/R]-L-V-H-V-V-E-[K/N]-L-[H/Q]-C-X-A-X$_{1-4}$-G-[I/V]-X-I-R-V. This group had been reported earlier [11] and includes AtMEK3 whose C-terminal extension is similar to the NTF2 domain, which is involved in the nuclear import of protein cargos [11, 12, 49, 67]. Although the NTF2 domain is found in most eukaryotic lineages, its fusion to a MEK protein has only been observed in Viridiplantae [11, 49]. AtMEK3 is the only member of group B MEK proteins that has been characterized; it plays important roles in jasmonate [68] and blue light responses [69]. To the best of our knowledge, no biological function has yet been ascribed to any other member of the

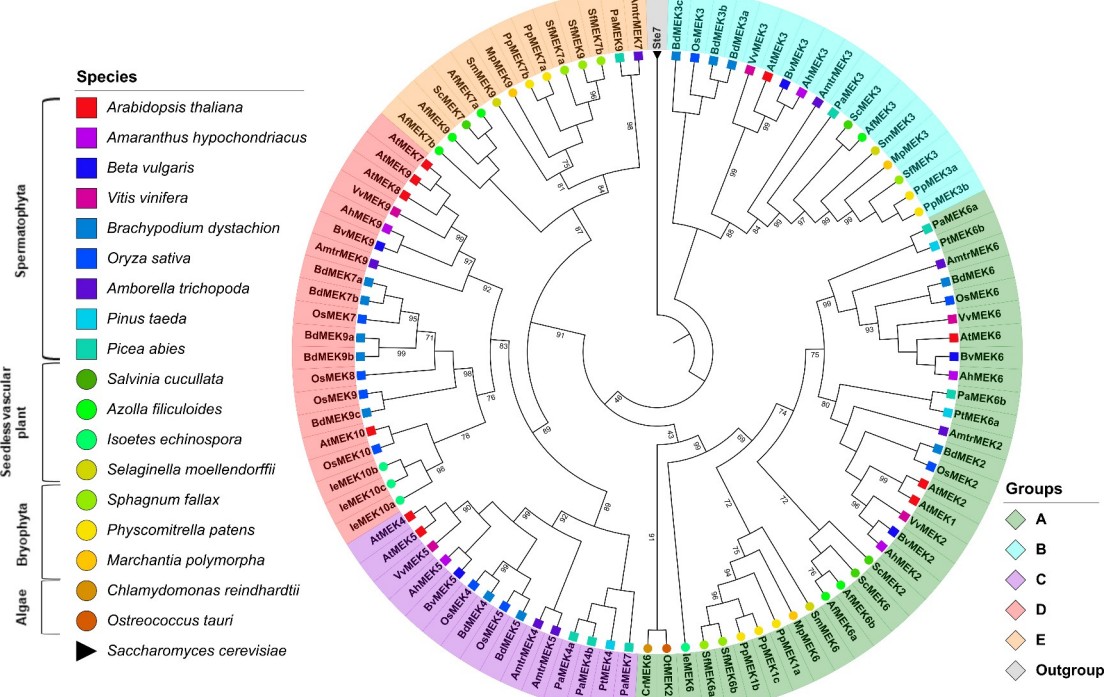

**Fig 4. Phylogenetic tree of MEK proteins in Viridiplantae.** Maximum-likelihood tree reconstructed from amino acid sequences. The tree was rooted with the Ste7 protein from Saccharomyces cerevisiae. Bootstrap values from 1,000 replicates are indicated at nodes with support values >90%. Each MEK group is indicated with a distinctive color. Different species are indicated with different shapes and colors.

group not even for the NTF2 domain *per se*; therefore, the biological processes in which group B MEKs are involved remain obscured. The presence of the NTF2 domain in all group B MEKs including sequences from angiosperms, gymnosperms, pteridophytes and bryophytes but no algae strongly suggests that the NTF2–MEK domain combination occurred after the divergence of embryophytes this is estimated to have occurred around 500 MYA [12]. Moreover, its conservation in taxa might reflect an evolutionary advantage in photosynthetic eukaryotes [49] although biochemical or functional characterization of group B MEKs are needed to corroborate this hypothesis.

The available information about groups C and D MEKs is scarce, but some members from this clade are involved in hormone signaling pathways. For example, AtMEK4/5 belongs to group C and mediates the ABA signaling pathways in response to biotic stress [34]. AtMEK7 and AtMEK9, belong to group D. They participate in the polar transport of auxins [70], and respond to ethylene [71], respectively. The novel group E proposed here contains sequences from bryophytes (*M. polymorpha*, *P. patens* and *S. fallax*), lycophytes (*S. moellendorffii*) pteridophytes (*A. filiculoides* and *S. cucullata*), gymnosperms (*P. abies*), and the early divergent angiosperm *A. trichopoda*. Interestingly, no group E MEKs were identified in monocot or dicotyledonous species, which suggests that these genes were lost prior to the Mesangiospermae origin, which diverged ~180 MYA [72]. Nonetheless, group E MEKs appears to be a sister clade to group C and D, which is well supported from bootstrap values (Fig 4).

**Analysis of motifs and domains conserved in the MEK family.** MEK proteins contain several conserved sequences to maintain the proper structure and function of these enzymes [73]. The MAPK-binding domain or docking-site (Table 5; S5 Table) is typically located towards the N-terminal end and its basic residues (lysine and arginine) interact with the acidic residues from the binding domain of MAPKs [64]. In addition, MEKs also contain an ATP-binding domain and the activation domain (Table 5; S5 Table) where serine/threonine residues are prone to phosphorylation and are involved in signal transduction [29]. Some proteins such as OsMEK7 or BdMEK7a have insertions within its activation domain, with one of serine/threonine residue absent or located around 3 to 5 residues upstream of the canonical position (S5 Table). Other proteins, such as ScMEK2, completely lack the MAPK-binding domain (S8 Fig; S5 Table). It is unclear whether these genes are expressed or if their gene products would exhibit MEK activity in MAP kinase cascades. If expressed, these proteins could represent neo or sub-functionalized versions arising from gene duplication events or even act as a noncatalytic scaffolding protein [12, 74].

**Table 5. Consensus sequences of the conserved domains in MEKs.**

| Domain | Consensus sequence |
| --- | --- |
| I | G-X-[S/A/N]-[G/S]-G-X-V-X-[K/L]-[V/A]-X-H-[K/R] |
| II | P-X-[V/L]-V-X-[C/F]-[H/Y]-$X_2$-[F/Y] |
| III | [L/M]-E-[Y/F]-M-D-X-G-S-L-[A/E] |
| IV | [A/S]-$X_6$-L-X-G-L-X-Y-L-H |
| V | V-G-T-$X_2$-Y-M-S-P-E-R-[I/F] |
| VI | [G/S/A]-D-[I/V]-W-S-[L/F]-G-[L/V]-$X_2$-L-E |
| VII | [F/A]-S-X-E-[F/L]-[R/C]-X-F-[I/V]-$X_2$-C |
| VIII | [S/T]-[A/V]-$X_2$-L-L-X-H-P-F-[I/V/L] |
| ATP-binding | H-K-$X_{3-8}$-A-L-K-$X_4$-[N/D]-X-[D/E/Q] |
| MAPK-binding | [V/I]-H-R-D-[I/L]-K-P-[S/A]-N-L-L |
| Activation | V-S-$X_5$-[S/T]-[M/L]-[D/G/A] |
| NTF2-binding | E-[K/R]-[L/I]-[V/I]-H-V-V-E-[K/N]-L-[Q/H]-C |

**Table 6. Predicted specific MAPK binding domain of plant MEKs.**

| Group | Sequence |
|---|---|
| A | H-X$_2$-[R/K]-[H/R]-[I/V]-I-H-R-D-[I/L]-K-P-S-N-L-L |
| B | H-X-V-R-H-L-V-H-R-D-I-K-P-A-N-[L/M]-L |
| C | H-X$_2$-R-X-I-V-H-R-D-X-K-P-[S/A]-N-L-L |
| D | H-X$_3$-K-I-V-H-R-D-I-K-P-X-N-L-L |
| E | H-K-X$_2$-[H/N]-K-I-V-H-R-D-I-K-P-S-N-L-L |

A scan of the retrieved MEK sequences with the MEME suite confirmed the existence of eight domains named I-VIII (Table 5; S5 Table). Domains I-IV are located prior the activation domain and are present in the vast majority of the MEK proteins analyzed except for PpMEK3a, PaMEK6, and IeMEK10a, which lack domain I and VvMEK9 that lacks domain II. VvMEK5 lacks domains I and II. BdMEK9c and CrMEK6 do not contain domain III. PaMEK10a lacks domains I-III, and AhMEK3 and ScMEK2 lack domain IV. In contrast, domains V-VIII are located towards the C-terminal end after the activation site. Domains V and VI are absent only in AfMEK6b; domain VII is not present in VvMEK2, VvMEK9, IeMK10b, and IeMEKc; and domain VIII is not found in VvMEK2, VvMEK9, AmtrMEK6, PtMEK6b, AtMPEK8, and PaMEK6b. Additionally, six other sequences lack at least two domains: PaMEK3 (V, VI, and VIII); PaMEK4a (VII and VIII); PAMEK4B (V and VII); PaMEK6a and IeMEK6 (VII and VIII); and IeMEK10 (VII and VIII) (S8 Fig; S5 Table). Nevertheless, the presence of these signature sequences could be used as a criterion to refine MEK homology searches or annotation.

**Identification of novel and distinctive domains for each group of MEKs.** MEK proteins include a MAPK binding domain with the consensus sequence [V/I]-H-R-D-[I/L]-K-P-[S/A]-N-L-L. This domain is involved in protein-protein interaction with its MAPK partner, and therefore is essential for MAPK signaling cascades [75]. A closer examination of this sequence reveals group-specific variations of the MAPK binding domain, which suggests specificity, at least to some level, in MEK selection towards its MAPK interactors prompting the hypothesis that each MEK group might be involved in specific biological processes (Table 6).

Six new motifs spanning 10–18 amino acid residues were identified in addition to the canonical MEK domains. Remarkably, these new motifs are group-specific and therefore can be used to quickly assign new MEK sequences to their respective ortholog group. These domains were named with consecutive cardinal numbers [76–81] (Fig 5; Table 7). With the exception of PaMEK6b, IeMEK6, PtMEK6a, and OtMEK2, all proteins from group A contain domain 9 that is located towards the N-ter. Proteins from group B contain domain 10 towards the N-terminal and domains 12 and 13 are towards the C-terminal. MEKs from the C and E group contain domains 11 and 14, respectively, located towards the C-terminal of the protein. MEK proteins from the D group do not contain any domains in addition to the canonical ones (Fig 5; S6 Table). A computational analysis of post-translational modifications showed that all the putative novel domains contain potential phosphorylation sites, which might suggest that these domains are functionally relevant and possibly regulate MEK activity; this hypothesis, however, requires experimental validation (S4 Table).

**Current knowledge on MEKK family.** Plant MEKK are the least studied component of MAPK signaling cascades; therefore, there is limited information for these proteins compared to the information available for MAPKs and MEKs [10, 16, 35, 82]. The *Arabidopsis* genome encodes at least 80 genes distributed in three kinase subfamilies whose orthologs in animals have been shown to have MEKK activity: MAPKKK (**MAPK**/ERK **k**inase **k**inase, 21 members), ZIK (**Z**R1-**i**nteracting **k**inase, 11 members) and RAF (**r**apid **a**cceleration **f**ibrosarcoma, 48

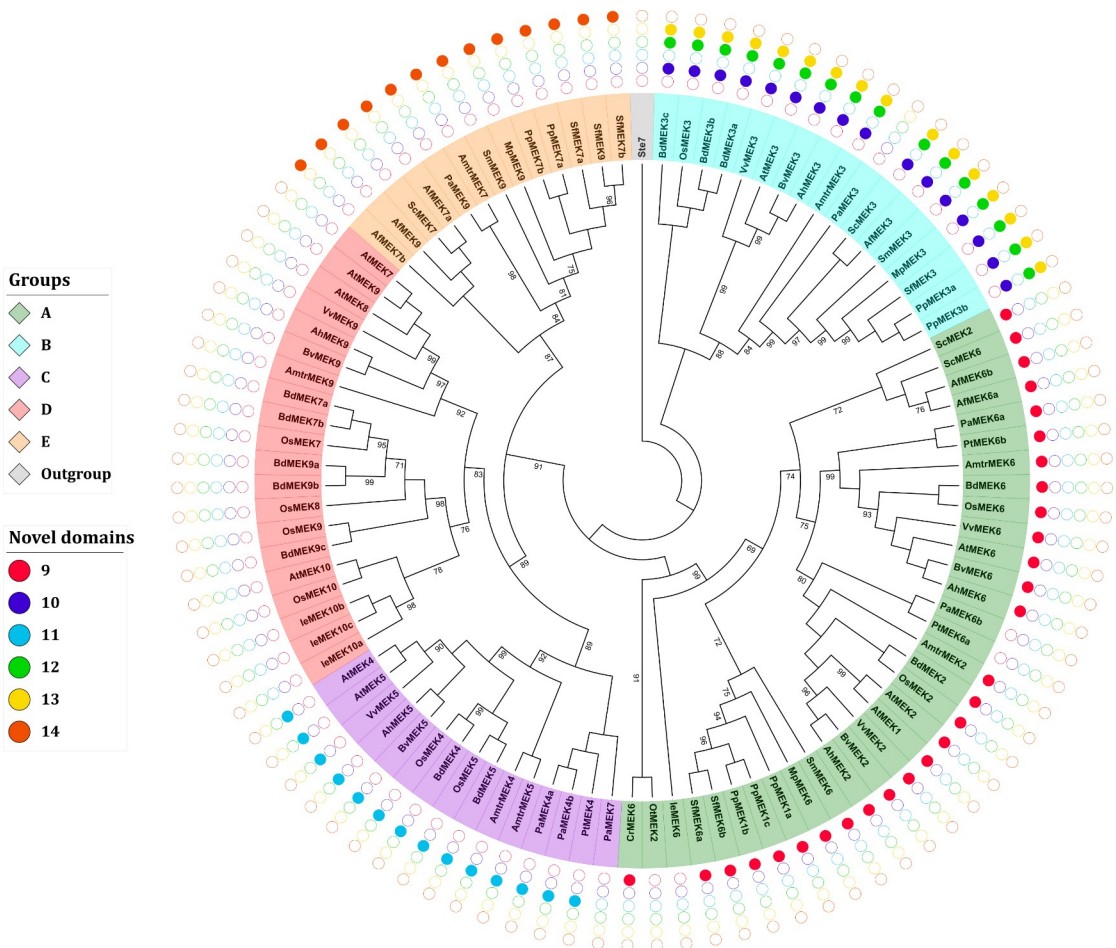

**Fig 5. Phylogenetic analysis and identification of novel MEK domains.** The analysis of novel domains was performed with the MEME suite [44]. The combinatorial distribution of the novel domains is group-specific and could facilitate future MEK identification and annotation. Sequence nomenclature follows a two-letter code to indicate genus and species (see text).

members) [16]. Kinase activity has been confirmed for several members of the MAPKKK subfamily in plants, but no evidence has been reported for members of the ZIK subfamily [16, 34, 36, 54]. CTR1 (Constitutive Triple Response 1) and EDR1 (Enhanced Disease Resistance 1), which belongs to the RAF subfamily and acts primarily as a negative regulators of MAP kinase modules [71, 83–85], but no functional characterization has been made for other RAF members.

Plant MPKKKs have a variable MAP kinase domain [14]. Most proteins from the MEKK group have a regulatory domain and a serine/threonine residue at the activation site [86]. The proteins belonging to the MEKK group have the consensus sequence G-[T/S]-P-X-[F/Y/W]-M-A-P-E-V in the kinase domain [19, 21]. Those from the ZIK group contain the sequence G-T-P-E-F-M-A-P-E-[L/V/M]-[Y/F/L] and are located towards the N-terminal end. Meanwhile, members of the RAF group have the sequence G-T-X-X-[W/Y]-M-A-P-E. In addition, most proteins from the RAF group have the kinase domain located towards the C-terminal end, while the N-terminal end contains a regulatory domain.

**Table 7. Consensus sequences of the novel domains discovered in MEKs.**

| Domain | Consensus sequence |
|---|---|
| 9 | [A/Q]-S-G-T-F-X-D-G-D-[L/I]-X-[L/V]-N-X$_2$-G |
| 10 | N-L-L-S-R-S-X$_3$-Y-N-[I/F]-N-E-X-G-[L/F] |
| 11 | L-P-[L/M]-P-X-R-X$_2$-D-X$_{2-4}$-S-L-A-V-P-L |
| 12 | M-L-[T/A]-[V/I]-H-Y-Y-[L/M]-L-F-[D/N]-G-X-D |
| 13 | G-[V/I]-X-I-R-S-G-S-F-[I/V]-V-G-X$_2$-F |
| 14 | [D/E]-P-P-X-P-P-X$_3$-[S/T]-P-X-F-X$_2$-F-I |

## Conclusions

MAPKs are among the oldest signal transduction pathways. In plants, they are involved in the regulation of various physiological processes such as hormone signaling [12, 15, 87] and respond to different types of stress [5, 14, 16, 17, 19]. Several members of the MEK family can act as convergence points of a great variety of signals and stimuli with an extraordinary high substrate specificity towards a MAPK. MAPKs on the other hand seem to function as divergence points [25]. Despite their involvement in several biological processes and the previous efforts to elucidate the evolutionary paths of MAPKs diversification in plants such convergence has been limited by the intrinsic properties of the MAPK signaling cascade components, i.e., its functional redundancy, high similarity at the amino acid level, gene family expansion and contraction in several plant taxa, and the availability of sequenced plant genomes.

Here, the phylogenetic relationships between MAPKs from 18 Viridiplantae species were analyzed. The analyzed species were selected to span the major Viridiplantae clades ranging from Chlorophyta, i.e., green algae, up to angiosperms. This allows for a comprehensive and genome wide exploration of MAPKs across Viridiplantae, and the consequent reconstruction of gene of gene phylogenies to assess the diversification of the MAPK signaling cascades in plants. The need to implement a heuristic nomenclature system has become more evident with the identification of a large number of MAPK and MEK family genes (189 and 98, respectively). In addition, the system proposed by Hamel *et al.* (2006) named MAPK genes according to their Arabidopsis orthologs and signature sequences that typify MAPK genes and has been extended. The proposed nomenclature might be used as a practical tool to aid in the identification of novel MAPK genes in additional plant genomes and even during genome annotation projects. The reconstructed ML trees exhibit a well resolved and supported topology; therefore, it is useful to recognize the most likely Arabidopsis ortholog of the retrieved MAPK sequences.

Orthology-based nomenclature systems provide functional insights for each MAPK clade [22, 27]; thus, some of the novel domains identified in this work could be correlated to substrate, activator, or inhibitor specificity as well as to protein-protein interactions. It is likely that some of the conclusions reached, or the number of identified sequences in some of the analyzed species, might need to be adjusted as new genome versions are updated or reannotated. Also, the inclusion of more genomes specifically from plant clades that are still under-represented such as lycophyte or gymnosperm might help to refine the reconstructed phylogenies. International collaborative projects such as the 10KP will overcome this gap in the near future [88].

## Supporting information

**S1 Fig. Multiple sequence alignment of representative MAPK proteins from selected plant species.** Twenty proteins representing the diversity of MAPKs in Viridiplantae were aligned using the MAFFT software [39] and visualized in Jalview [40]. The numerals I to XI indicate

the conserved (canonical) kinase domains; TXY and CD labels indicate the activation site and docking domain, respectively.
(TIF)

**S2 Fig. Distribution of the different activation domains in MAPK proteins.** Most proteins from the A, B, and C group contain TEY activation motif, while proteins from group D contain the TDY activation motif. Group E exhibits a high variability of activation motifs especially in early divergent plant species.
(TIF)

**S3 Fig. Biochemical characteristics of MAPK groups.** Boxplots showing a) molecular weight, b) protein length, and c) theorical isoelectric point of MAPK proteins according to their group. The boxplot indicates the median line as well as first and third quartiles. Outliers that are 1.5 above the upper quartile or below the lower quartile are indicated as points.
(TIF)

**S4 Fig. Phylogenetic and canonical domain analysis of MAPKs.** The identification of the canonical MAPK domains was performed with the MEME suite [44]. Sequence nomenclature follows a two-letter code to indicate genus and species (see text).
(TIF)

**S5 Fig. Molecular structure of plant MAPKs showing putative localization of the novel domains.** The protein structures were obtained from homology modelling [47] and visualized in PyMol (The PyMOL Molecular Graphics System, Version 2.0 Schrödinger, LLC.). Each domain is presented as a sequence logo, obtained from the multiple sequence alignment of all the retrieved sequences [46], and highlighted in the protein struure with a distinctive color.
(TIF)

**S6 Fig. Biochemical characteristics of MEK groups.** Boxplots showing a) molecular weight, b) protein length, and c) theorical isoelectric point of MEK proteins according to their group. The boxplot indicates the median line, first, and third quartiles. Outliers that are 1.5× above the upper quartile or below the lower quartile are indicated as points.
(TIF)

**S7 Fig. Multiple sequence alignment of representative MEK proteins from different plant species.** Twenty proteins representing the diversity of MEKs across Viridiplantae were aligned using the MAFFT software [39] and visualized in Jalview [40]. The numerals I to VIII represent the canonical domains; the ATP-binding and MAPK-binding domain are also indicated.
(TIF)

**S8 Fig. Phylogenetic and canonical domains analysis of MEKs.** The identification of the canonical MEK domains was performed with the MEME suite [44]. Sequence nomenclature follows a two-letter code to indicate the genus and species (see text).
(TIF)

**S1 Table. List of the MAPK and MEK proteins identified in the genomes of selected species.**
(XLSX)

**S2 Table. Sequences of the canonical domains in the MAPK family.**
(XLSX)

**S3 Table. Sequences of the novel domains identified in the MAPK family.**
(XLSX)

**S4 Table. Computational prediction of phosphorylation sites of the novel domains identified in the MAPK and MEK families.**
(XLSX)

**S5 Table. Sequences of the canonical domains in the MEK family.**
(XLSX)

**S6 Table. Sequences of the novel domains identified in the MEK family.**
(XLSX)

**S1 File. Multiple sequence alignment of MAPK proteins.**
(TXT)

**S2 File. Multiple sequence alignment of MEK proteins.**
(TXT)

**S3 File. Newick format of the structure of the phylogenetic tree of MAPK proteins.**
(TXT)

**S4 File. Newick format of the structure of the phylogenetic tree of MEK proteins.**
(TXT)

## Acknowledgments

The authors thank Gustavo Delgado-Prudencio and Adrian Martínez-Santana (graduate students in the Instituto de Biotecnología-UNAM) for their insightful suggestions and help with phylogenetic analysis. The technical assistance from MTI Juan Manuel Hurtado-Ramírez and MEM David Santiago Castañeda-Carreón (Unidad de Cómputo del Instituto de Biotecnología-UNAM) is also greatly acknowledged. Finally, we want to thank the Plos Publishing Fee Assistance Program (https://plos.org/publish/fees) for their support.

## Author Contributions

**Conceptualization:** Ángel Arturo Guevara-García.

**Data curation:** José Manuel González-Coronel, Gustavo Rodríguez-Alonso.

**Formal analysis:** José Manuel González-Coronel, Gustavo Rodríguez-Alonso.

**Funding acquisition:** Ángel Arturo Guevara-García.

**Investigation:** José Manuel González-Coronel, Gustavo Rodríguez-Alonso.

**Supervision:** Gustavo Rodríguez-Alonso, Ángel Arturo Guevara-García.

**Writing – original draft:** José Manuel González-Coronel, Gustavo Rodríguez-Alonso, Ángel Arturo Guevara-García.

**Writing – review & editing:** José Manuel González-Coronel, Gustavo Rodríguez-Alonso, Ángel Arturo Guevara-García.

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
