## [Decision Letter · Decision Letter 0]

18 Feb 2021

PONE-D-21-00271

Phylogenetic study of the members of the MAPK and. MEK families across Viridiplanta

PLOS ONE

Dear Dr. Guevara-García,

Thank you for submitting your manuscript to PLOS ONE. After careful consideration, we feel that it has merit but does not fully meet PLOS ONE’s publication criteria as it currently stands. Therefore, we invite you to submit a revised version of the manuscript that addresses the points raised during the review process.

We look forward to receiving your revised manuscript.

Kind regards,

Ruslan Kalendar, PhD

Academic Editor

PLOS ONE

Reviewers' comments:

Reviewer's Responses to Questions

**Comments to the Author**

1. Is the manuscript technically sound, and do the data support the conclusions?

Reviewer #1: Yes

Reviewer #2: Partly

2. Has the statistical analysis been performed appropriately and rigorously? 

Reviewer #1: Yes

Reviewer #2: Yes

3. Have the authors made all data underlying the findings in their manuscript fully available?

Reviewer #1: Yes

Reviewer #2: Yes

4. Is the manuscript presented in an intelligible fashion and written in standard English?

Reviewer #1: Yes

Reviewer #2: No

5. Review Comments to the Author

Reviewer #1: 

The manuscript and the analysis performed is technically sound and mostly well written.

However, the manuscript is a bit too long and is difficult for readers and reviewers to understand the specific scientific and methodological contributions made in this specific study.

The figures provided in the PDF are also completely not legible and kind of defeats the purpose. The authors should work on improving the resolution of the figures when submitting for peer review.

Reviewer #2: 

The comments and suggestions for improving the current manuscript have been inserted in the attached file. The english language of the manuscript should be improved.

---

## [Author Response · Author response to Decision Letter 0]

6 Apr 2021

In this version 2.0, we considered all the editorial comments kindly raised by the reviewers. Heeding the suggestions of reviewer #1, wee shortened the manuscript a bit and tried to put more emphasis on our contributions, we hope this has helped make our writing more understandable. With this idea in mind, Figures 3 and 6 from the previous version, are now included in the supplementary material as Figures S4 and S8, respectively. In the other hand, we fully agree with the reviewer that resolution of the figures in pdf format is very low. All the Figures were prepared according to PACE specifications and we hope that in addition to the pdf files generated by the submission system, the reviewer will have access to download and to see tiff files with adequate resolution. Regarding the comments of reviewer 2, we first want to apologize for the inappropriate use of language. English is certainly not our mother tongue, however, personally (corresponding author) I am a frequent reviewer of articles submitted for publication in international journals, including Plos One. In any case, attending the reviewer's comment, we resorted to a professional scientific manuscript editing service (American Manuscript Editors; Attached Certificate). We hope this version 2.0 better written, more understandable and easier to read. Besides, we attended, if not all, most of the suggestions that reviewers indicated in the attached file.

In this version 2.0 submitted, all changes made and content additions are written in blue, and the parts that we propose to be deleted are written in red and crossed out.

In the file “Revised Manuscript with Track Changes”, all the modifications made to the original manuscript are marked and highlighted, including those suggested by American Manuscript Editor. Additionally, to make it easier to follow up on suggestions made by reviewers, here we include a list of the modifications made.

List of modifications

Header (title, affiliations and addresses)

Page 1, Lines 4-7; The address of the affiliation institution was modified and postal code was included.

Page 1 Line 9; The Email address of the corresponding author was updated, as it was recently changed.

Abstract

Page 2, Line 16; The following paragraph was deleted:

MAPK´s families are ubiquitous in eukaryotes and each one is constituted by a variable number of proteins in different species. In the Viridiplantae Líneage, they have been implicated in the control of multiple processes, including stress responses and developmental programs.

Introduction

Page 3, Line 45; The following sentence was removed:

Plant genomes contain a large number of genes encoding protein kinases, which constitute a protein superfamily.

Page 3, Line 52; The following paragraph was relocated to the "discussion" section in line 529:

MAPKs are among the oldest signal transduction pathways and in plants are involved in the regulation of various physiological processes such as the detection of hormones and the response to different types of stress.

Page 3, Line 63; The following paragraph was relocated to the "discussion" section in line 532:

Various members of the MEK family can act as convergence points of a great variety of signals and stimuli, with an extraordinary high substrate specificity towards a MAPK. MAPKs, on the other hand, seem to function as divergence points.

Page 4, Line 82; The following paragraph was deleted:

The availability and analyses of plant genome sequences have facilitated the identification of various members of the MAP kinase family in numerous plant species, contributing to their evolutionary study, as well as, the elucidation of the signaling mechanisms involved in various types of responses.

Results and Discussion

Page 8, Line 181; Table 1 title was replaced 

Page 10, Line 223; the title of Fig. 1 was replaced

Page 10, Line 240; the title of Fig. 2 was replaced 

Page 14, Line 323; Fig. 3 was changed to the Supplementary Materials section and renamed S4 Fig. 

Page 16, Lines 388 & 392; Fig. 4 was renamed Fig. 3 and the title was replaced, respectively.

Page 16, Line 397; Table 3 title was replaced.

Page 17, Line 427; Table 4 title was replaced.

Page 19, Lines 451 & 452; Fig. 5 was renamed as Fig. 4.

Page 21, Line 492; Fig. 6 was changed to the Supplementary Materials section and renamed S8 Fig.

Page 21, Line 500; Table 5 title was replaced.

Page 22, Line 523; Table 6 title was replaced.

Page 23, Line 528; Fig. 7 was renamed as Fig. 5.

Conclussions

Page 24, Line 564; The following sentence removed:

MAPK signaling cascades are conserved and widespread in eukaryotic organisms, where they are crucial in signal transduction in response to biotic, abiotic, and developmental cues

Page 25, Line 577; The following paragraph was removed:

The analyzed species were selected to span the major Viridiplantae clades, ranging from Chlorophyta, i.e., green algae, up to angiosperms. Previous approaches to elucidate the evolutionary relationships of plant MAPKs were biased towards angiosperms, as the first plant sequenced genomes belonged to species from this clade 16. Here we overcome this issue by including unicellular chlorophyta (Ostreococcus tauri and Chlamydomonas reindhardtii), bryophytes (Marchantia polymorpha, Physcomitrella patens, and Sphagnum fallax); the genome and transcriptome of Selaginella moellendorffii and Isoetes tegetiformes, respectively, both belonging to the lycophyte clade 74,97; the recently released genomes of pteridophyte (Azolla filiculoides and Salvinia cucullata) 68; and the gymnosperm genomes (Picea abies and Pinus taeda). As for the angiosperms we include basal species 

(A. trichopoda), and both monocots and dicots. This allows for a comprehensive and genome-wide exploration of MAPKs across Viridiplantae; and the consequent reconstruction of gene phylogenies to assess the diversification of the MAPK signaling cascades in plants. Our approach led to the identification of.

We have done the best of our effort to improve our manuscript trying to satisfactorily address all reviewer comments, which undoubtedly improved our research and report. Hopefully, you may find it acceptable for publication in Plos One. 

Finally, in case our report is accepted for publication in Plos One, we want to make public our gratitude to the Plos Publication Fee Assistance Program (https://plos.org/publish/fees) for approved support. Therefore, a paragraph about it, was added to the acknowledgments section.

Respectfully, and on behalf of all authors,

Dr. Ángel Arturo Guevara-Garcia

arturo.guevara@ibt.unam.mx

Instituto de Biotecnología (http://www.ibt.unam.mx)

---

## [Editor Report · Decision Letter 1]

12 Apr 2021

A phylogenetic study of the members of the MAPK and MEK families across Viridiplantae.

PONE-D-21-00271R1

Dear Dr. Guevara-García,

We’re pleased to inform you that your manuscript has been judged scientifically suitable for publication and will be formally accepted for publication once it meets all outstanding technical requirements.

Kind regards,

Ruslan Kalendar, PhD

Academic Editor

PLOS ONE

---

## [Editor Report · Acceptance letter]

14 Apr 2021

PONE-D-21-00271R1 

A phylogenetic study of the members of the MAPK and MEK families across Viridiplantae. 

Dear Dr. Guevara-García:

I'm pleased to inform you that your manuscript has been deemed suitable for publication in PLOS ONE. Congratulations! Your manuscript is now with our production department. 

Kind regards, 

on behalf of

Prof. Ruslan Kalendar 

Academic Editor

PLOS ONE